# Ensemble-Based Online Machine Learning Algorithms for Network Intrusion Detection Systems Using Streaming Data

**Nathan Martindale**, **Muhammad Ismail \*** and **Douglas A. Talbert**

Department of Computer Science, College of Engineering, Tennessee Tech University,
Cookeville, TN 38505, USA; namartinda42@students.tntech.edu (N.M.); dtalbert@tntech.edu (D.A.T.)
\* Correspondence: mismail@tntech.edu

**Abstract:** As new cyberattacks are launched against systems and networks on a daily basis, the ability for network intrusion detection systems to operate efficiently in the big data era has become critically important, particularly as more low-power Internet-of-Things (IoT) devices enter the market. This has motivated research in applying machine learning algorithms that can operate on streams of data, trained online or "live" on only a small amount of data kept in memory at a time, as opposed to the more classical approaches that are trained solely offline on all of the data at once. In this context, one important concept from machine learning for improving detection performance is the idea of "ensembles", where a collection of machine learning algorithms are combined to compensate for their individual limitations and produce an overall superior algorithm. Unfortunately, existing research lacks proper performance comparison between homogeneous and heterogeneous online ensembles. Hence, this paper investigates several homogeneous and heterogeneous ensembles, proposes three novel online heterogeneous ensembles for intrusion detection, and compares their performance accuracy, run-time complexity, and response to concept drifts. Out of the proposed novel online ensembles, the heterogeneous ensemble consisting of an adaptive random forest of Hoeffding Trees combined with a Hoeffding Adaptive Tree performed the best, by dealing with concept drift in the most effective way. While this scheme is less accurate than a larger size adaptive random forest, it offered a marginally better run-time, which is beneficial for online training.

**Keywords:** network intrusion detection; stream data; online learning

## 1. Introduction

In an increasingly technological society, our reliance on networks of systems has dramatically exploded. This manifests itself in the vast adoption of Internet-of-Things (IoTs). As we continue to add more and more devices and sensors into these networks to improve the quality of our lives, we also increase the threat surface and potential for attacks. Hence, there is a need to appropriately secure these networks in order to protect our data. Given how much data flows through these networks on a regular basis, manual analysis of traffic logs is inadequate for detecting malicious intrusions into the network. This creates an incentive to develop better automated systems for analyzing this traffic to judge whether it is benign or malicious. These automated systems are known as Network Intrusion Detection Systems (NIDS).

As technologies such as the IoT further develop, more and more research is being conducted to improve the performance and efficiency of NIDS [1]. These performance improvements are especially important as more low-power devices are involved. Machine learning has a useful crossover into the domain of cybersecurity and has been applied to many problems such as spam, phishing, and malware



detection [2] as well as to data theft and network attacks [3]. Through techniques such as anomaly detection and traffic classification, NIDS can be automated and benefit from the developments in machine learning to enhance detection performance. In this context, online learning and ensembles can contribute to the development of efficient NIDS. Specifically, online learning supports on-the-fly training for data streaming, rather than relying on a static dataset. This is useful in domains where live update and detection must take place. Furthermore, ensembles provide a method for combining multiple instances of machine learning models that may complement each other, and thus provide a more accurate overall system performance.

Ensembling is a powerful approach in machine learning where multiple separately trained models are combined in various ways [4]. An example of how models can be combined is the common approach of taking a simple majority vote—given an input, each model in the ensemble makes a prediction, and the majority predicted answer is the final predicted classification. The power of ensembling is based on the idea of a diverse set of models, allowing for the strengths of some models to balance out the weaknesses of other models, resulting in better overall predictions [5].

The experiments in this work focus primarily on using streaming data. Rather than conventional approaches such as in offline learning where all of the data is loaded and trained on at once, referred to as "one-shot data mining" [6], streaming based approaches can take in a continuous stream of realtime data and only keep a certain amount of that data in memory at a time before discarding it. This is a form of online learning where models generally need to update on the fly, in order to deal with changing data distributions referred to as concept drift [7]. The need for this ability to handle streaming data has increased as the rate of data collection in many areas has exploded—in some instances, using the total amount of data collected in a regular static approach would be infeasible [6].

The idea of concept drift is important to discuss and understand for this topic. As mentioned, concept drift refers to a change in the distribution of the data over time, but this can manifest in several different ways, as discussed in [7,8]. Sudden concept drift occurs when the distribution immediately jumps to another distribution. A gradual concept drift more slowly slides from one distribution to another. A recurring concept drift might have a cyclic pattern in shifting between multiple distributions. Concept drift can present a problem when an algorithm learns a function that models one distribution but not what it drifts to—the potential function underlying the data is no longer the same, which jeopardizes the performance of NIDS. Strong stream-based learning algorithms need to be able to detect such a drift and update its underlying model.

A major technical challenge in NIDS is that while ensembles, or combinations of many instances of machine learning algorithms, are likely to lead to better classification results, they also tend to require more processing power. There is a tradeoff between accuracy, run-time, and response to concept drifts. As a result, while designing efficient NIDS, it may be necessary to determine where different sets of techniques fall with respect to such a tradeoff. In the literature, there has not been a study that explicitly compares the different types of ensembles for NIDS on streaming data, and this work seeks to fill that gap and make recommendations regarding the best ensemble technique that balances these tradeoffs. Hence, the contributions of this work can be summarized as follows:

- We explore and compare several stand-alone detectors along with homogeneous and heterogeneous ensembles for malicious traffic detection using the KDDCup99 dataset [9]. Specifically, we investigate stand-alone detectors such as K-nearest neighbor (K-NN), Support Vector Machine (SVM), Hoeffding Adaptive Tree (HAT), and Adaptive Random Forest (ARF). In addition, we investigate homogeneous ensembles based on HAT and ARF. Finally, we propose three heterogeneous ensembles based on HAT + ARF, SVM + HAT, and SVM + ARF.
- We run experiments to investigate the performance of NIDS with streaming data involving multiple protocols, namely, 600,000 HTTP connections and 100,000 SMTP connections. We analyze both the accuracy and run-time of the aforementioned NIDS against these traffic types.

- We investigate the performance of the aforementioned NIDS when concept drift is considered. Through experimental results, we show that the heterogeneous ensemble of HAT + ARF handles concept drift better than the other detectors.

The rest of this paper is organized as follows: Section 2 reviews the related work. Section 3 discusses the dataset preparation, stand-alone detectors, along with homogeneous and heterogeneous ensembles that are investigated in this paper. Section 4 presents experimental results and discusses performance evaluation metrics. Finally, conclusions are drawn in Section 5.

## 2. Related Work

There are two different types of ensembles that are relevant to this work: homogeneous and heterogeneous. Homogeneous ensembles are composed of multiple instances of the same type of model. Random forests are a commonly used example of this. A random forest is a homogeneous ensemble, or an ensemble made up of many individual decision trees. Heterogeneous ensembles, in contrast, are made up of different types of classifiers. Ensembling different algorithms such as a neural network, an SVM, and a decision tree would be an example of a heterogeneous ensemble. While heterogeneous ensembles are not inherently or universally better, it is easier to get greater diversity from them, potentially leading to higher accuracy [10]. Several works have explored applying ensembles to NIDS and shown that these ensemble approaches, usually random forests, can be highly effective. In [11], the authors search for the optimal number of decision trees to include in the forest, and explore the performance/efficiency tradeoffs for different sizes while run in Apache Spark. In [12], the authors compare random forests with individual algorithms such as Naive Bayes, SVM, K-NN, and a decision tree, and it is found that the random forest offers the highest precision and accuracy. In [13], several stand-alone algorithms such as decision trees and multi-layer perceptrons, and different ensembles for NIDS such as random forests and boosting are compared in terms of accuracy and response-time on a raspberry pi. It is found that decision trees and extreme gradient boosting offer the best performance/time tradeoffs. All aforementioned works, however, do not consider stream data.

Works like [14] have explored anomaly detection in stream data. Special algorithms designed for stream data have been used, such as Robust Random Cut Forests in [15]. Tying in ensembles from above, several other papers have looked at comparing ensemble approaches with single (stand-alone) algorithms for NIDS or other security problems. For instance, [16] compares a K-NN, HAT, ARF, and SVM on stream data. Various types of decision trees ensembled with bagging and/or boosting have been explored in [17,18]. Autoencoders have been used both ensembled together [19] as well as individually compared with random forests [20]. Also, Ref. [21] offers an example of a heterogeneous ensemble of a K-NN, SVM, and multi-layer perceptron in order to detect malicious URLs.

While there are many works evaluating the use of ensemble approaches for streaming NIDS, none explicitly compare the performance of heterogeneous and homogeneous ensembles. This comparison is what we explore in this work. With the rising need for processing power due to the deployment of more complex algorithms and given the increasing number of low-powered IoT devices, it is critical to analyze and find a balance in tradeoffs between run-time performance, anomaly-detection accuracy, and response to concept drifts. Analyzing different kinds of ensembles could help us to find a good performance-accuracy balance, especially when concept drift is considered.

## 3. Methodology

Our work aims to compare the performance of a set of heterogeneous and homogenous ensembles in terms of accuracy, run-time, and response to concept drifts. In our approach, we use the massive online analysis (MOA) framework [22] to stream an NIDS dataset into the various machine learning algorithms. Each ensemble or stand-alone algorithm runs through a form of online supervised learning where each sample is tested and then trained on in sequence. We then look at the average accuracy and total CPU run-time for each approach.

*3.1. Data Preparation*

The dataset used in this paper is the KDDCup99 [9], a commonly used dataset for NIDS and anomaly detection. This dataset is a large collection of tcpdump data, where each row is a connection record, referring to the aggregate collection of a sequence of packets that are all associated with a single connection. It contains several different types of labeled simulated intrusions, including denial-of-service (DoS) attacks, probing attacks, etc., amounting to a total of 24 different attack types in the training data. The connection types are diverse and include multiple protocols and services, including HTTP, SMTP, telnet, and more. In total there are 41 different features describing each connection. In our experiments, we consider nearly 600,000 HTTP connections and 100,000 SMTP connections. The reason for selecting these subsets is that they present distinct concept drifts within them. Further, the two connection types were not only tested individually but were also combined so that the HTTP data started at the end of the SMTP data. This creates a sudden and dramatic concept drift that we have intentionally added, in combination with the concept drift inherent to the separate protocol sets themselves. In all of our experiments, a methodology similar to [17] was used for constructing and turning the dataset into a data stream. Specifically, we use the MOA framework [22], which handles processing a collection of offline data as a datastream by feeding in a few points to the algorithm at a time.

*3.2. Stand-Alone Algorithms*

In this subsection, we briefly describe the different algorithms used throughout this work. All experiments were conducted using MOA, which presents ready implementations for each of the algorithms tested. The MOA framework is a system written in Java, designed for running and exploring data stream mining algorithms. The MOA is intended to emulate WEKA [23] for stream learning, which offers a similar work flow as WEKA and can ingest data from similar .arff sources. This framework includes many common evaluation metrics used in online and stream learning, allows for rapid experimentation and testing with a graphical interface to manipulate and manage algorithm parameters, and simplifies the process of setting up data for streaming it into a model. All algorithms considered in this work represent supervised learning approaches that are trained and evaluated with MOAs "EvaluatePrequential" task. This takes each incoming sample, tests on it, and then trains the model with that new sample. It should be noted that unless explicitly mentioned, all algorithms were run with the default parameters specified by MOA.

3.2.1. K-Nearest Neighbors

K-NN [24] is a basic supervised classification algorithm. It operates based on the principle that data with similar characteristics or features tend to also have the same classification. For any new piece of input data to be classified, the $K$ other known points that are most similar to the input piece will "vote" based on their classification. The majority vote will then be used as the classification for the new data point.

3.2.2. Support Vector Machines

SVMs [25] learn a dividing line between two classes that maximizes the distance between the nearest point to the line of each class. This requires the data to be linearly separable, which may require the data to be mapped into a higher dimensional space. The distance between the support vectors, the two parallel vectors that define the boundaries of the nearest classification point of that class, is what the algorithm attempts to maximize.

3.2.3. Hoeffding Adaptive Trees

An HT is a type of decision tree designed for incremental learning or streaming data [26]. It is particularly known for its efficiency and ability to generate trees similar to a tree that would result from

training on the entire dataset at once in a non-streaming fashion. HAT is a variant of these proposed in [27]. Adaptive trees use adaptive windowing (ADWIN) of recent data to detect high levels of change that indicate a need to create a new or altered tree. As a result, HATs tend to handle concept drift better than a regular HT.

### 3.3. Homogeneous Ensembles: Adaptive Random Forests

Random forests are collections or ensembles of multiple decision trees, trained slightly differently for instance by injecting randomness into which features are used at splits. ARFs apply a similar concept in a stream data mining context, by using decision trees capable of handling incremental learning, and were originally proposed in [28]. The ARFs in our experiments use HTs as the base learner.

### 3.4. Proposed Heterogeneous Ensembles

Ensembles of models can frequently outperform individual models [21]. The experiments conducted in this paper are designed to compare the performance of homogeneous ensembles, made up of the same base learning algorithm, with heterogeneous ensembles, made up of different algorithms. In particular, we compare the performance of ensembles made up of K-NN, HAT, ARF, and SVM. The aforementioned algorithms were chosen as they offer the highest detection performance on streaming data, as highlighted in [16]. It should be noted that ARFs and SVMs tend to be high performing algorithms, with the best ability to adapt to concept drift, according to the experiments of [16]. Furthermore, the experiments in [16] revealed that HAT offers the best results versus its offline variant but is heavily affected by concept drifts in the data.

Hence, in our work, we propose heterogeneous ensembles that consist of a "stable" learner (i.e., SVM or ARF) and a HAT model such that the periods of higher detection accuracy from the HAT could potentially be maintained while the more quickly adapting learner balances or cancels out the HAT's inability to quickly adjust to concept drifts. Hence, in one ensemble detector, a HAT is combined with an ARF, as illustrated in Figure 1a, and in another ensemble an SVM with a HAT is considered, as illustrated in Figure 1b. Finally, a third ensemble that consists of an SVM and an ARF, as illustrated in Figure 1c, is investigated to determine the impact of combining two different stable learners together. The third detector offers a benefit that SVMs and random forests tend to have lower false positives [20]. All algorithms are ensembled together using a weighted majority classifier.

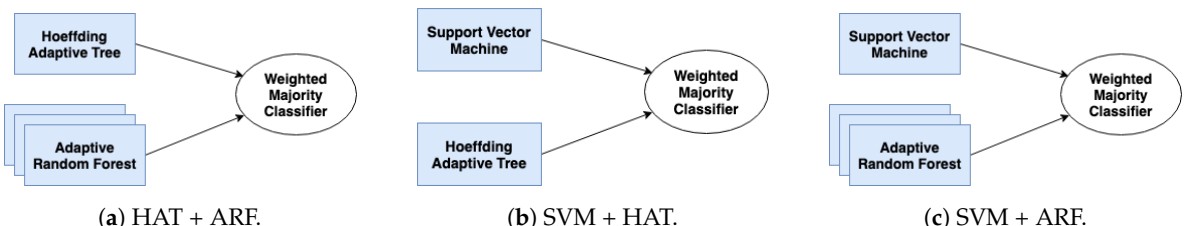

(**a**) HAT + ARF.　　　　　(**b**) SVM + HAT.　　　　　(**c**) SVM + ARF.

**Figure 1.** Proposed heterogeneous ensembles. Each subfigure shows the listed algorithms combined with a weighted majority classifier.

Given a stream of live data representing traffic connections within a network, the goal of these detectors is to learn on the fly to distinguish between anomalous/malicious and normal/benign traffic. Specifically, since it is stream-based, the algorithms only retain a small amount of the data at a time, and must incrementally train or update as new data comes in, rather than batch training it on all of the data in an offline setting. Since concept drift is a major challenge in any online learning system, particularly in one dealing with network traffic, it is important that the solution is able to adapt to changes in the data distribution quickly and effectively. The important considerations are: (1) a balanced false positive to false negative ratio, and (2) not requiring undue computational resources to run and train online. The challenges for online learning include the difficulty in accurately assessing

or evaluating the approaches, which require different metrics such as prequential area under the curve (AUC) evaluation, which is discussed further in Section 4.1.

## 4. Experimental Results

In total, 11 algorithms were run against three different KDDCup99 subsets, the results of which are presented in the tables below. In each table, the average AUC of each run is shown with the total run-time (in seconds) beneath it. The ARF was run with the default settings, composed of 10 HTs. When labeled as "ARF (20)", it was run with 20 trees. The K-NN experiments used $K = 10$, and both of the boosted approaches were made up of five models.

### 4.1. Evaluation Metrics

All algorithms were evaluated with a metric called *prequential evaluation* [16]. Prequential evaluation is based on AUC, i.e., the area under the receiver operating characteristic (ROC) curve. The ROC curve plots the true positive rate versus the false positive rate, as the threshold of a binary classifier is changed. The prequential AUC evaluation technique based on this was proposed in [29], which operates by incrementally calculating the AUC of a sliding window range of previous data. This version of the AUC metric was designed specifically for streaming data. It should be highlighted that both [16,17] use this AUC as the primary metric for comparison, as this is a more reliable indicator when dealing with imbalanced datasets. In this study, both prequential AUC as well as the total run-time of the algorithm are collected and compared. Time analysis is critical, as in a live system where resources are limited, it is important to consider the necessary computation time needed to run the various approaches.

### 4.2. Stand-Alone Models

This subsection presents the performance evaluation results for stand-alone detectors. All experiments were run in the experimenter tab in MOA, and in each table, the values reported are directly from MOA's results and summary sections, using the AUC and evaluation time (CPU run-time in seconds) metrics.

Table 1 shows the performance of each algorithm when run individually. In general, the HT performed the best, and SVM had the fastest run-times at the expense of accuracy. It is worth noting that the K-NN model exhibits significantly longer run-time compared with other tested approaches, with a run-time of nearly 10 min compared to the next longest run-time of less than 10 s.

**Table 1.** Stand-alone model results. Bolded values are the best results between the four different model types.

|  | K-NN | SVM | HT | HAT |
|---|---|---|---|---|
| HTTP | $0.95 \pm 0.13$ | $0.96 \pm 0.13$ | $\mathbf{0.99 \pm 0.06}$ | $0.97 \pm 0.10$ |
|  | 516.07 s | **4.60 s** | 6.59 s | 7.00 s |
| SMTP | $0.87 \pm 0.18$ | $0.80 \pm 0.28$ | $0.90 \pm 0.15$ | $\mathbf{0.91 \pm 0.14}$ |
|  | 89.77 s | 1.16 s | **0.87 s** | 1.38 s |
| Combined | $0.94 \pm 0.14$ | $0.86 \pm 0.25$ | $\mathbf{0.99 \pm 0.07}$ | $0.96 \pm 0.11$ |
|  | 593.32 s | **5.27 s** | 9.51 s | 8.74 s |

### 4.3. Ensemble Algorithms

This subsection presents the performance evaluation results for both homogeneous and heterogeneous ensembles.

4.3.1. Accuracy and Run-Time Results

Table 2 compares different homogeneous ensemble approaches, where each ensemble is made up of multiple instances of the same base learner. All these ensembles present high prequential AUC, and specifically the high performance of ARF (20) demonstrates that adding more trees into the forest can improve the performance, at the cost of computation run-time. Twice as many trees in the forest corresponds to roughly twice as much run-time, as would be expected. A similarly intuitive result when comparing Table 2 with Table 1 is that the run-times of the ensemble algorithms are nearly always notably higher than the individual algorithms, aside from the K-NN experiments. All of the algorithms presented in Table 2 offer quite high accuracy, and so it should be noted that in order to better compare these algorithms, we should focus more on the run-time as well as investigate how the algorithm reacts to concept drift.

**Table 2.** Homogeneous ensembles.

|          | BoostHT | BoostHAT | ARF | ARF (20) |
|----------|---------|----------|-----|----------|
| HTTP | $0.98 \pm 0.08$ | $0.98 \pm 0.09$ | $0.98 \pm 0.10$ | $\mathbf{0.99 \pm 0.07}$ |
|          | 25.28 s | **15.62 s** | 17.57 s | 29.88 s |
| SMTP | $0.96 \pm 0.09$ | $0.97 \pm 0.06$ | $0.97 \pm 0.08$ | $\mathbf{0.98 \pm 0.08}$ |
|          | 3.48 s | 4.74 s | **3.26 s** | 5.53 s |
| Combined | $0.99 \pm 0.05$ | $0.99 \pm 0.06$ | $0.98 \pm 0.09$ | $\mathbf{1.00 \pm 0.03}$ |
|          | 62.62 s | 41.41 s | **32.72 s** | 67.61 s |

Table 3 presents experimental results for the proposed heterogeneous ensembles, consisting of combinations of distinct base learners. Both of the ensembles containing an ARF offer similar performance, although their run-times are significantly greater than the SVM and HAT combination. Interestingly, the accuracy of the SVM and HAT combination is only notably lower than the ARF ensembles on the subsets including SMTP traffic. In terms of accuracy, the HAT and ARF combination offers the highest accuracy, with the correspondingly highest run-times.

**Table 3.** Heterogeneous ensembles.

|          | HAT + ARF | SVM + HAT | SVM + ARF |
|----------|-----------|-----------|-----------|
| HTTP | $0.98 \pm 0.10$ | $0.97 \pm 0.10$ | $\mathbf{0.98 \pm 0.10}$ |
|          | 25.2 s | **8.45 s** | 19.99 s |
| SMTP | $0.98 \pm 0.05$ | $0.93 \pm 0.14$ | $\mathbf{0.98 \pm 0.08}$ |
|          | 4.56 s | **1.61 s** | 4.17 s |
| Combined | $\mathbf{0.98 \pm 0.09}$ | $0.94 \pm 0.15$ | $0.97 \pm 0.11$ |
|          | 55.59 s | **11.16 s** | 44.39 s |

The following remarks can be made based on the observations from Tables 1–3:

- While stand-alone models offer comparable accuracy results to ensemble models on HTTP and combined connections, ensemble models offer up to 8% improvement in accuracy for SMTP connections. Such improvement is achieved, however, at the cost of increasing the run-time, roughly by three times (except for the K-NN model).
- Since the models presented in Tables 1–3 offer close accuracy performance, further experiments are required to investigate if the model's performance might differ as we plot graphs of the AUC over time, which corresponds to the number of instances the algorithm has trained on. This is presented next.

4.3.2. Concept Drift Results

We compare the prequential AUC performance over time for: (a) all three heterogeneous ensembles and (b) heterogeneous ensembles versus stand-alone models.

Comparison of Proposed Heterogeneous Ensembles

Figure 2 compares the three proposed experimental ensembles' performance on the HTTP subset. The large dips in AUC result from the concept drifts. While the three approaches struggle at several (and almost same) points, the SVM + HAT (orange) fails in several additional places, and the HAT + ARF (in green) is resilient to at least one major change that the SVM + ARF (blue) does not correctly handle, just after 400,000 instances into the stream. This implies that the HAT + ARF approach is likely more robust than the others.

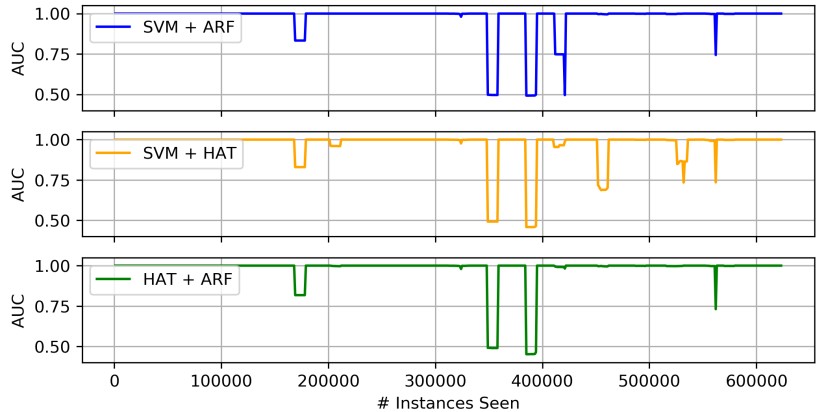

**Figure 2.** Comparison of heterogeneous ensemble performances on the HTTP data subset.

Figure 3 shows the same comparison when ensembles are run on the SMTP data. Similarly to Figure 2, these results show that while both ARF ensembles have comparable average AUC performances, the HAT + ARF ensemble is marginally more resilient to the concept drift in the data than the SVM + ARF ensemble.

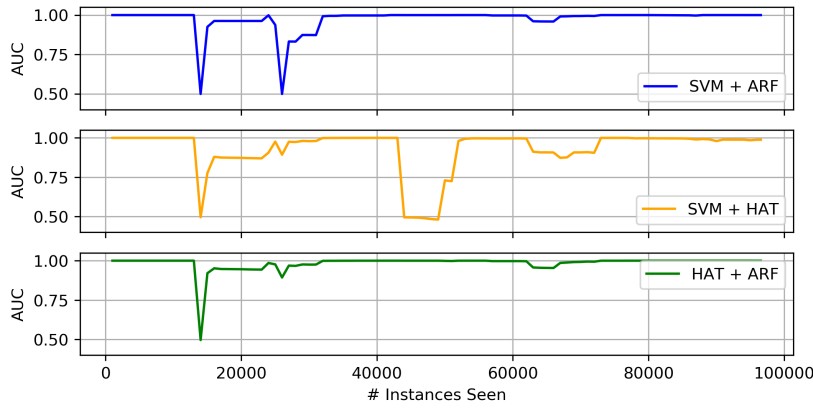

**Figure 3.** Comparison of heterogeneous ensemble performances on the SMTP data subset.

Figure 4 compares the performance of heterogeneous ensembles on the combined SMTP and HTTP dataset. Interestingly, in the latter portion of the graph (on the HTTP data), while many of the same high error/concept drifts as in Figure 2 occur, the AUC drops notably further specifically for the combined SVM + HAT. This seems to indicate that the SVM and HAT models do not recover as well from starting the learning process with significantly different data.

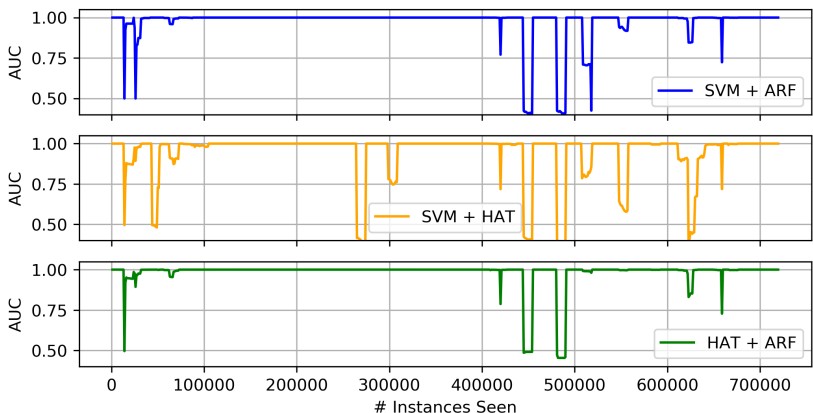

**Figure 4.** Comparison of heterogeneous ensemble performances on combined dataset.

Comparison of Heterogeneous Ensembles and Their Individual Components

Figures 5 and 6 plots AUC over time for the HAT + ARF (blue) ensemble versus the individual HAT (green) and ARF (orange) models. The ARF and HAT are impacted by different concept drifts in the data, and the fewer/lesser dips in AUC of the HAT + ARF show that the ensemble of these two is able to combine their individual strengths into a stronger overall model.

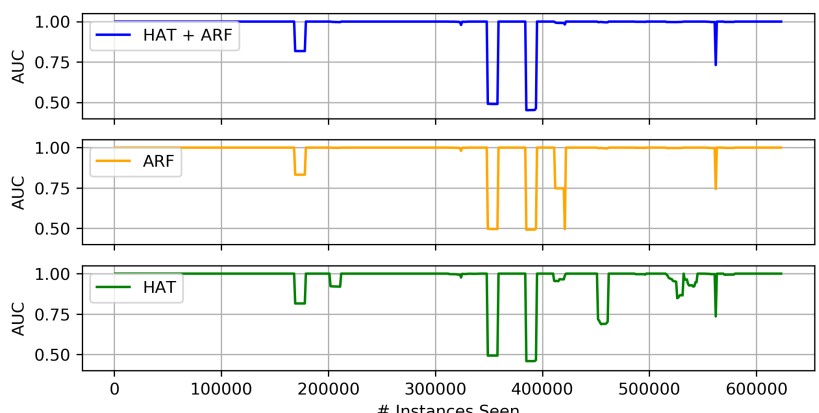

**Figure 5.** Comparison of HAT + ARF ensemble and their individual component performances on HTTP.

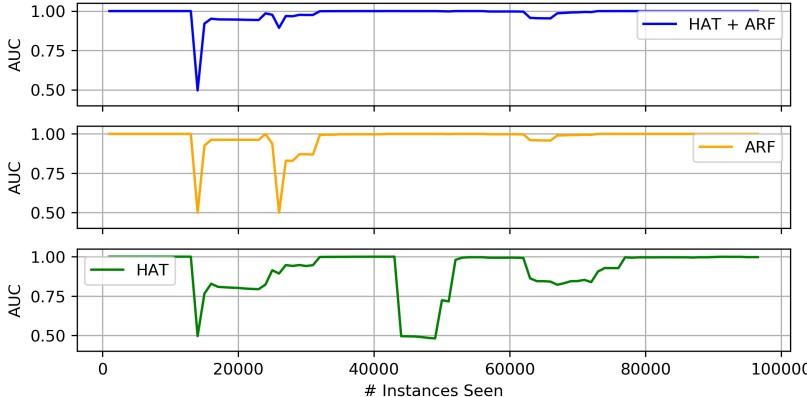

**Figure 6.** Comparison of Hoeffding Adaptive Tree (HAT) + Adaptive Random Forest (ARF) ensemble and their individual component performances on SMTP.

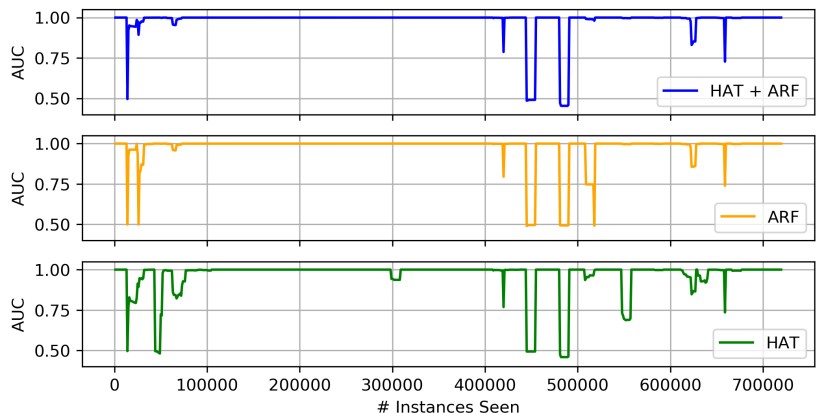

**Figure 7.** Comparison of HAT + ARF ensemble and their individual component performances on combined data.

A similar pattern is observed in Figure 7, as in Figure 5. Interestingly, when comparing this result with Figure 4, we find that the HAT alone does not have many of the same concept drift problems as when combined with the SVM, further supporting Table 1 that the SVM is the weaker approach of the algorithms tested.

Finally, in Figures 8 and 9, the best heterogeneous ensemble, HAT + ARF (blue), is compared with simply increasing the number of trees in a homogeneous ARF (orange) ensemble. Comparing Tables 2 and 3, a larger ARF has marginally better average AUC than the HAT + ARF ensemble. However, there is no clear performance advantage between these two in terms of concept drift—in the HTTP instances. While there is a major drop in AUC at one concept drift point with HAT + ARF, there are many small points where the HAT + ARF ensemble handles concept drifts better than the ARF. Similarly, in the SMTP data, while there is one point after 60,000 instances that the HAT + ARF drops to a small degree where the ARF does not, just after 20,000 the HAT + ARF ensemble still handles the concept drift much better than the homogeneous ARF ensemble. It should also be noted that the HAT + ARF ensemble runs in slightly less time than larger ARF.

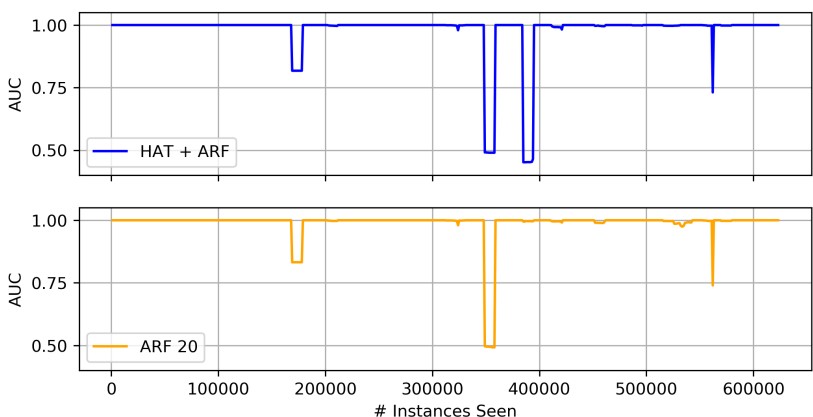

**Figure 8.** Performance comparison of heterogeneous (HAT + ARF) and homogeneous (ARF-20) ensembles on HTTP.

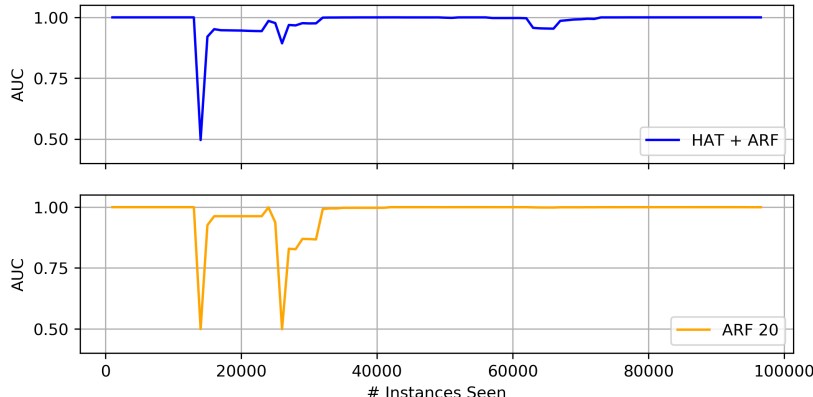

**Figure 9.** Performance comparison of heterogeneous (HAT + ARF) and homogeneous (ARF) ensembles on SMTP.

The following concluding remarks can be made based on the experimental results:

- Almost all of the ensemble techniques, both homogeneous and heterogeneous, lead to higher average AUC than the base (stand-alone) algorithms (i.e., decision trees, SVM, and K-NN). In most cases this comes with at least a slight cost in terms of run-time, and hence, computation resources.
- Of the three proposed heterogeneous ensembles, the HAT + ARF offers the highest accuracy performance. In particular, this approach is obviously better than the two individual algorithms (i.e., HAT and ARF) when looking at the AUC over time, where it recovers or sufficiently handles concept drift in the data stream.
- When comparing HAT + ARF with a larger ARF, close performance is observed, and the larger ARF even performs better in some instances, but with a slightly higher run-time.

*4.4. Discussion*

It is important to note that the ARF algorithm uses regular HTs as the base learner. Hence, the HAT + ARF approach is combining a HAT with a regular HT. The adaptive tree differs in that it utilizes ADWIN, which automatically expands or shrinks the window of data used based on the amount of change present within it, which explains the higher resilience to concept drift than the ARF or HT learners. The ensemble of a HAT and ARF of HTs may be effective in that it has access to both adaptive windowing and non-adaptive windowing over recent data. This may allow greater flexibility in learning how to handle different kinds of concept drift.

As noted, most ensembles, while having a greater run-time, performed better than the stand-alone algorithms with the exception of K-NNs. The K-NN had poor run-time performance in all instances, and so this would be a technique to avoid on live implementations. This high run-time may be due to the need to continuously calculate the distance or similarity metrics between the new sample and every other sample still in memory, rather than the relatively fewer operations needed to evaluate a decision tree.

Breaking the results of the approaches down into the AUC over time graphs also highlights and reinforces an interesting point that average AUC by itself does not necessarily indicate how an algorithm responds to concept drift. Despite the fact that the larger ARF had similar overall average performance to the combined HAT + ARF, they responded slightly differently to different concept drifts present in the data, as evidenced in Figures 8 and 9.

## 5. Conclusions

This paper explored the performance and run-time tradeoffs of various machine learning approaches to network intrusion detection. Specifically, all tested algorithms were trained online, with streaming traffic connection data, meaning it can only train on a small portion of the total data at

a time. Several individual algorithms were tested, as well as a few homogeneous ensemble approaches, along with three proposed heterogeneous ensembles. Overall, the ensembles performed better than the individual base learners, but with a generally higher run-time. One of the proposed heterogeneous ensembles, HAT + ARF, presented a comparable accuracy performance with the best homogeneous ensemble model (the ARF) but with slightly lower run-time.

In future work, more investigations will be carried out to determine how the ARF's structure can be optimized to achieve the best performance with a minimal run-time. The ensemble offering the best performance is composed of a single HAT with 10 HTs, but it is possible that having different counts of the respective base learners could achieve a higher performance than a larger HAT by itself. For example, combining five HATs with an ARF of five HTs may allow a better balance or weighting between the different methods of windowing data. Similarly, some of the related works discussed using one-class anomaly detection algorithms as part of an ensemble, and it is possible bringing in an autoencoder combined with smaller ARF's could further improve performance.

**Author Contributions:** Conceptualization, N.M.; methodology, N.M.; supervision, M.I.; writing—original draft, N.M.; writing—review and editing, M.I. and D.A.T. All authors have read and agreed to the published version of the manuscript.

**Funding:** This research received no external funding.

**Conflicts of Interest:** The authors declare no conflict of interest.

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
