# Peer review of "Ensemble-Based Online Machine Learning Algorithms for Network Intrusion Detection Systems Using Streaming Data"

_information, doi:10.3390/info11060315_

Round 1
Reviewer 1 Report
The motivation should be explained more clearly.
The Introduction is written as a summary, which is not what an Introduction should be.
The authors should extend the introduction section.
Related work section should be separated from the introduction and written as section 2 not sub-section from the introduction.
The proposed method is inadequately described. Better start by providing the reader with a high level picture of the problem.
There is no analysis of the extracted results and no discussion.
Separate results and discussion. Results are facts, discussion is your interpretation of these facts.
Author Response
The motivation should be explained more clearly. The Introduction is written as a summary, which is not what an Introduction should be. The authors should extend the introduction section. Related work section should be separated from the introduction and written as section 2 not sub-section from the introduction.
We have now included two separate sections as: Introduction and Related Works sections. Additional text describing the motivation and challenges of the subject area have been added to the Introduction (highlighted in red starting at line 64 in the revised manuscript).
The proposed method is inadequately described. Better start by providing the reader with a high-level picture of the problem.
We have added a better overall description of our approach at the beginning of the methodology section (highlighted in red starting at line 122 in the revised manuscript).
There is no analysis of the extracted results and no discussion. Separate results and discussion. Results are facts, discussion is your interpretation of these facts.
A discussion section has been added (Section 4.4, highlighted in red starting at line 332 in Section 4.4 of the revised manuscript).
Reviewer 2 Report
The authors of this paper explored the performance and run-time trade-offs of various machine learning approaches to network intrusion detection. Specifically, all tested algorithms were trained online, with streaming traffic connection data, meaning it can only train on a small portion of the total data at a time. Several individual algorithms were tested, as well as a few homogeneous ensemble approaches, along with three proposed heterogeneous ensembles. Overall, the ensembles performed better than the individual base learners, but with a generally higher run-time.
Revision:
-Authors would elaborate on how the ARF’s structure can be optimized to achieve the best performance with a minimal run-time. The ensemble offering the best performance is composed of a single HAT with 10 HT’s, but it is possible that having an ensemble of two smaller ARF’s, one with HAT as a base learner and one with HT as a base learner could achieve a higher performance than a larger HAT, and with a lower run-time.
-Which is the confidence interval in the experimental results?
Author Response
Authors would elaborate on how the ARF’s structure can be optimized to achieve the best performance with a minimal run-time. The ensemble offering the best performance is composed of a single HAT with 10 HT’s, but it is possible that having an ensemble of two smaller ARF’s, one with HAT as a base learner and one with HT as a base learner could achieve a higher performance than a larger HAT, and with a lower run-time.
The idea here is that potentially by changing the balance between the two different windowing techniques present in an adaptive random forest of hoeffding trees and hoeffding adaptive trees, it might alter how well it can train/respond to different concept drifts, and this would be an interesting question to explore. We have elaborated more on this point as a topic of future research (highlighted in red starting on line 362 in the revised manuscript).
Which is the confidence interval in the experimental results?
The confidence intervals, as reported by the MOA framework, are listed along with the average AUC values in Tables 1, 2, and 3 (as a value).
Reviewer 3 Report
The article reflects some substantial work showing structured thinking and good elaboration on the results. However, the “minor issues” described below should be addressed:
- The authors should specify how did they obtain the values in tables 1, 2, and 3.
- The authors should give a brief description of the simulation platform that they have used.
- For the comparison figures, the authors should homogenize the scale on the “Y” axis as it is different, sometimes it is based on (.6, .8, and 1) and sometimes (.5, .75, and 1).
- The authors should make the text in the blue boxes and the oval forms in figure 1 more legible.
- On line 23, the tense of the verb “grow,” or the whole sentence should be revised.
Author Response
The authors should specify how did they obtain the values in tables 1, 2, and 3.
We have elaborated more on how we used the MOA software to collect the results (highlighted in red starting on line 232 in the revised manuscript).
The authors should give a brief description of the simulation platform that they have used.
We have added further details about the MOA software (highlighted in red starting on line 151 in the revised manuscript).
For the comparison figures, the authors should homogenize the scale on the “Y” axis as it is different, sometimes it is based on (.6, .8, and 1) and sometimes (.5, .75, and 1).
We have adjusted all tick marks in the graphs to reflect the same range, with ticks at (.5, .75, and 1).
The authors should make the text in the blue boxes and the oval forms in figure 1 more legible.
The font size in the diagrams was increased and set to bold for clarity.
On line 23, the tense of the verb “grow,” or the whole sentence should be revised.
We have revised the whole sentence to make it cleaner (highlighted in red on line 22-23 in the revised manuscript).
Reviewer 4 Report
In this paper, the authors propose heterogeneous ensembles that consist of a “stable” learner (i.e., SVM or ARF) and a HAT model such that the periods of higher detection accuracy from the HAT could potentially be maintained while the more quickly adapting learner balances or cancels out the HAT’s inability to quickly adjust to concept drifts.
Here are my comments.
1) The title of this paper can be changed as “An ensemble-based Online Machine Learning Algorithms for Network Intrusion Detection Systems Using Streaming Data” because Ensemble methods is a machine learning technique that combines several base models in order to produce one optimal predictive model.
2) Please compare your paper with the following paper [*]. Then, the authors emphasize the efficiency of Online Machine Learning Algorithms.
[*] Verma, A., Ranga, V. Machine Learning-Based Intrusion Detection Systems for IoT Applications. Wireless Pers Commun 111, 2287–2310 (2020). https://doi.org/10.1007/s11277-019-06986-8.
3) Please discuss more ensemble-based online machine learning algorithms. Here is simple when the authors use the exiting methods and combine them, such as a HAT is combined with an ARF, as illustrated in Figure 1a, and in another ensemble, an SVM with a HAT is considered, as illustrated in Figure 1b, and consists of an SVM and an ARF, as illustrated in Figure 1c.
What is your contribution/novelty in this paper? Please explain clearly.
Author Response
The title of this paper can be changed as “An ensemble-based Online Machine Learning Algorithms for Network Intrusion Detection Systems Using Streaming Data” because Ensemble methods is a machine learning technique that combines several base models in order to produce one optimal predictive model.
We have revised the title accordingly.
Please compare your paper with the following paper [*]. Then, the authors emphasize the efficiency of Online Machine Learning Algorithms.
[*] Verma, A., Ranga, V. Machine Learning-Based Intrusion Detection Systems for IoT Applications. Wireless Pers Commun 111, 2287–2310 (2020). https://doi.org/10.1007/s11277-019-06986-8.
We have added this paper to our related works section (starting on line 109). This paper is similar in approach, and we differ in that we focus explicitly on homogeneous versus heterogeneous ensembles for streaming data. We have clarified this in the revised manuscript (highlighted in red starting line 102).
Please discuss more ensemble-based online machine learning algorithms. Here is simple when the authors use the exiting methods and combine them, such as a HAT is combined with an ARF, as illustrated in Figure 1a, and in another ensemble, an SVM with a HAT is considered, as illustrated in Figure 1b, and consists of an SVM and an ARF, as illustrated in Figure 1c. What is your contribution/novelty in this paper? Please explain clearly.
The trade-off between time complexity, accuracy, and response to concept drifts in homogeneous and heterogeneous ensembles has not been explicitly addressed before in literature. Our work aims to shed some light on this topic and highlights the best ensembles that balances such a trade-off. In the revise manuscript, we have given a more explicit statement of our contribution near the end of the Introduction (highlighted in red starting on line 64).
Round 2
Reviewer 1 Report
The authors have addressed all my previous comments.
The paper can be accepted now
Reviewer 4 Report
My comments were addressed.